# Changes in microRNAs during Storage and Processing of Breast Milk

**DOI:** 10.3390/metabo13020139

**Published:** 2023-01-17

**Authors:** Jun Hwan Kim, Ki-Uk Kim, Hyeyoung Min, Eun Sun Lee, In Seok Lim, Jeonglyn Song, Insoo Kang, Dae Yong Yi

**Affiliations:** 1Department of Pediatrics, Chung-Ang University Hospital, Seoul 06973, Republic of Korea; 2College of Medicine, Chung-Ang University, Seoul 06972, Republic of Korea; 3College of Pharmacy, Chung-Ang University, Seoul 06972, Republic of Korea; 4Chung-Ang University Industry Academic Cooperation Foundation, Seoul 06972, Republic of Korea; 5Departments of Internal Medicine, Section of Rheumatology, Allergy & Immunology, Yale University School of Medicine, S525C TAC, 300 Cedar Street, New Haven, CT 06520, USA

**Keywords:** human milk, microRNAs, freezing

## Abstract

Human breast milk (HBM) is the ideal source of nutrients for infants and is rich in microRNA (miRNA). In recent years, expressed breast milk feeding rather than direct breastfeeding has become increasingly prevalent for various reasons. Expressed HBM requires storage and processing, which can cause various changes in the ingredients. We investigated how the miRNAs in HBM change due to processes often used in real life. HBM samples collected from 10 participants were each divided into seven groups according to the storage temperature, thawing method, and storage period. In addition, we analyzed the miRNA changes in each group. The number of microRNAs that showed significant expression was not large compared to the thousands of miRNAs contained in breast milk. Therefore, it is difficult to suggest that the various storage and thawing processes have a great influence on the overall expression of miRNA. However, a short-term refrigeration storage method revealed little change in nutrients compared to other storage and thawing methods. Taking all factors into consideration, short-term refrigeration is recommended to minimize changes in the composition or function of breast milk.

## 1. Introduction

Human breast milk (HBM) contains complex proteins, lipids, and carbohydrates, and is the ideal source of nutrients for infants. Additionally, it contains antibodies involved in the human immune response and oligosaccharides involved in shaping the intestinal microbiota and supporting health, among numerous other biologically active compounds [1,2]. Breastfeeding provides economic and medical benefits to mothers, but also improves short-term health outcomes in the infant, such as lowering the risk of infection, diarrhea, sudden infant death syndrome, and other childhood diseases and conditions, and long-term health benefits, such as lowering the risk of diabetes, obesity, and asthma after infancy [3,4]. According to these findings, the World Health Organization (WHO), United Nations International Children’s Emergency Fund (UNICEF), and the American Academy of Pediatrics (AAP) encourage governments to create an environment supportive of breastfeeding or the provision of breast milk and recommend exclusive breastfeeding until six months after birth. As the importance of breastfeeding has emerged, interest and education in breastfeeding have been increasing in Korea over the past 20 years, and the breastfeeding rate has also been steadily increasing [5,6]. Expressed breast milk feeding rather than direct breastfeeding has become increasingly prevalent, especially in some developed countries, because of various maternal factors and for practical reasons. Expressed breast milk undergoes various storage and processing procedures, which can change the composition, ingredients, and functions, compromising the nutritional quality of breast milk [7]. Therefore, it has become necessary to clarify such changes in the macronutrient and immunological components [8].

Breast milk is particularly diverse and rich in microRNA (miRNA). Accordingly, there is much interest in the role of miRNA in breast milk, particularly how miRNAs contained in breast milk are nutritionally absorbed through the infant’s digestive tract and benefit the infant, as well as their role as epigenetic regulators of gene expression in infants [9,10]. miRNAs are small (approximately 22 nucleotides), single-stranded, non-expressing RNAs that form a complex with proteins to engage in RNA silencing or bind to target messenger RNAs (mRNAs) to direct post-transcriptional silencing. By suppressing the expression of their target genes, miRNAs are regulators of cell and tissue development, differentiation, proliferation, and metabolism [11]. miRNAs regulate at least 60% of human mRNAs, and some of the tens of thousands of miRNAs are known to be related to cancer and other diseases, while ongoing research efforts are directed at examining the feasibility of miRNAs as biomarkers for the diagnosis, treatment, and recurrence prediction of diseases [12]. In breast milk, miRNAs are speculated to function as immune protectors and developmental regulators between infants and nursing mothers [13]. Milk-derived miRNAs may not only serve as a fingerprint of the mother’s health but also of potential outcomes to the health of the infant receiving the milk [9,10,11,12]. Mir-181 and mir-155 are associated with the differentiation in B cells, and mir-17 and mir-92 affect the differentiation and maturation of B and T cells. HBM-derived miRNAs also regulate the proliferation of intestinal epithelial cells, have a preventive effect on atopy, and are key regulators of milk lipid metabolism.

To perform their functions, miRNAs must remain stable. Despite the rise in expressed breast milk feeding, there has been little analysis of changes in miRNA in expressed breast milk. Our research seeks to determine how the miRNA in expressed breast milk is affected by various storage and handling conditions often used in real life. Based on these studies, optimal breastfeeding management guidelines can be presented, and an environment can be provided to enhance the convenience of nursing mothers and promote the best health conditions for infants.

## 2. Methods

### 2.1. Collection of Breast Milk and Storage and Processing

Human breast milk was collected at the Chung-Ang University Breastfeeding Research Institute (Seoul, Korea) for current lactating mothers who agreed to participate in the study. There was no particular restriction on the research participants because our focus was to compare only the change according to storage and thawing with initially collected HBM and confirm the stability of miRNAs. Breast milk was collected in a fresh state within a day of milking and was collected after refrigerating for less than 24 h at home except when delivered immediately after the milking. The donated breast milk of 40 to 50 cc was divided into seven aliquots (samples) immediately after collection, and each sample was treated, stored, and thawed as follows (Figure 1): Sample 1: initial miRNA analysis; Sample 2: microwave treatment (700 W, 60 s); Sample 3: stored at room temperature (10–20 °C) for one week; Sample 4: refrigerated (4 °C) for one week; Sample 5: stored frozen for one week and then thawed in a bottle warmer (300 W, 5 min; Philips, Amsterdam, The Netherlands); Sample 6: stored frozen for one week and then thawed by heating in a microwave (700 W, 60 s); Sample 7: stored frozen for four weeks, and then thawed by heating in a bottle warmer (300 W, 5 min).

The seven samples were compared for their changes in miRNAs according to microwave treatment (Sample 1, Sample 2), storage condition (Sample 1, Sample 3, Sample 4), thawing method (Sample 1, Sample 5, Sample 6), and duration of frozen storage (Sample 1, Sample 6, Sample 7).

### 2.2. Human Breast Milk Fractionation and miRNA Isolation

miRNA analysis of each sample was performed according to the breast milk collection date and storage period. To prepare the nonfat skim milk fraction, human breast milk samples were centrifuged (3000× *g*, 4 °C, 10 min) twice, and fat, cells, and debris were removed. The skim milk samples were centrifuged (12,000× *g*, 4 °C, 20 min) and sequentially filtered through 0.8, 0.45, and 0.22 μm syringe filters (Sartorius AG, Göttingen, Germany) to remove residual fat and cell debris. miRNAs of skim milk were extracted by acid-phenol/chloroform separation combined with column-based filtration using the mirVana miRNA Isolation Kit (Invitrogen, Waltham, MA, USA). The concentration and purity (260:280 ratio) of miRNAs were analyzed using a NanoDrop™ 1000 spectrophotometer (Thermo Scientific, Waltham, MA, USA). Aliquots (>20 ng RNA) of the miRNA samples were used for small RNA sequencing (RNA-seq).

### 2.3. Library Preparation and Sequencing

The library was constructed using a NEBNext Multiplex Small RNA Library Prep Kit (New England BioLabs, Inc., Ipswich, MA, USA) according to the manufacturer’s instructions. Briefly, for library construction, total RNA from each sample was used to ligate the adaptors, and cDNA was then synthesized using reverse-transcriptase with adaptor-specific primers. PCR was performed for library amplification, and the libraries were cleaned up using a QIAquick PCR Purification Kit (Qiagen, Inc., Hilden, Germany) and polyacrylamide gel electrophoresis gel. The yield and size distribution of the small RNA libraries were assessed with an Agilent High Sensitivity DNA Assay on an Agilent 2100 Bioanalyzer instrument (Agilent Technologies, Inc., Santa Clara, CA, USA). High-throughput sequences were produced by a NextSeq 500 system using 75 bp single-end sequencing (Illumina, San Diego, CA, USA).

### 2.4. Data Analysis

Sequence reads were mapped by the Bowtie2 software tool to obtain a bam file. Mature miRNA sequences were used as a reference for mapping. Read counts mapped to mature miRNA sequences were extracted from the alignment file using BEDtools v2.25.0 and a bioconductor using the R statistical programming language. Read counts were used for determining the expression level of miRNAs. The counts per million-trimmed mean of M-values (CPM-TMM) normalization method was used to compare samples. For the miRNA target study, miRWalk 2.0 was performed. Functional gene classification was performed by DIANA. In addition, differentially expressed gene analysis was used to identify miRNAs with a significant expression. We found miRNA with a normalized RC (log2) value of 4 or higher, with the expression of miRNA increasing or decreasing by more than twofold on a subgroup basis and the *p*-value of 0.05 in the *t*-test.

## 3. Results

### 3.1. Participant Information

Breast milk was collected 10 times from nine lactating mothers (Table 1). Participants 3 and 6 are the same lactating mother who participated in different periods of breastfeeding. Each collection time was postpartum 12 and 146 days. The average age of the lactating mother was 34.4 years, and the baby’s postpartum period was 84.6 days at the sample collection date. The average gestational age for babies was 35 weeks and one day, and the average birth weight was 2.59 kg. In the HBM sample of 10 participants, the difference in expression was confirmed according to the storage and thawing method of 2588 microRNAs.

### 3.2. Changes in miRNA According to Microwave Treatment of Breast Milk

When comparing the expression level of miRNAs between the control (Sample 1) and microwave treatment (Sample 2), two miRNAs showed significant changes in all participants (hsa-miR-24-3p and hsa-miR-27a-3p). However, there was no tendency for increased and decreased gene expression, according to the participant (Table 2, Figure 2).

### 3.3. Changes in miRNA According to the Storage Conditions of Breast Milk

#### 3.3.1. Changes in miRNA According to One Week of Storage at Room Temperature

When we compared the expression level of miRNAs between the control (Sample 1) and one week of storage at room temperature (Sample 3), we saw that four miRNAs showed significant changes in all participants (hsa-miR-193b-5p, hsa-miR-365a-3p, hsa-miR-365b-3p, and hsa-miR-378a-3p). The expression level of hsa-miR-378a-3p was significantly decreased in all participants. Furthermore, hsa-miR-365a-3p and hsa-miR-365b-3p were significantly decreased in all participants except for Participant 6 (Table 2, Figure 2).

#### 3.3.2. Changes in miRNA According to One Week of Refrigerated Storage

When we compared the expression level of miRNAs between the control (Sample 1) and one week of refrigerated storage (Sample 4), we saw that no miRNAs showed significant changes in all participants (Table 2, Figure 2).

#### 3.3.3. Changes in miRNA According to Bottle Warmer Thawing after 1 Week of Frozen Storage

When we compared the expression level of miRNAs between the control (Sample 1) and bottle warmer thawing after one week of frozen storage (Sample 5), we saw that two miRNAs showed significant changes in all participants (hsa-miR-205-5p and hsa-miR-3182). hsa-miR-3182 was significantly decreased in all participants except for Participant 8 (Table 2, Figure 2).

### 3.4. Changes in miRNA According to the Thawing Method of Breast Milk

#### 3.4.1. Changes in miRNA According to Bottle Warmer Thawing after 1 Week of Frozen Storage

To explore the change in miRNA of breast milk according to the thawing method, breast milk was exposed to different thawing methods (bottle warmer and microwave treatment) after one week of frozen storage. Again, when we compared the expression level of miRNAs between the control (Sample 1) and bottle warmer thawing after one week of frozen storage (Sample 5), we saw that two miRNAs showed significant changes in all participants (hsa-miR-205-5p and hsa-miR-3182). hsa-miR-3182 was significantly decreased in all participants except for Participant 8 (Table 2, Figure 2).

#### 3.4.2. Changes in miRNA According to Microwave Thawing after 1 Week of Frozen Storage

When we compared the expression level of miRNAs between the control (Sample 1) and microwave thawing after one week of frozen storage (Sample 6), six miRNAs we saw significant changes in all participants (hsa-miR-10b-5p, hsa-miR-205-5p, hsa-miR-486-5p, hsa-miR-3960, hsa-miR-24-3p, and hsa-miR-378a-3p). hsa-miR-3960 was significantly increased in all participants except for Participant 4, and hsa-miR-24-3p was significantly decreased in all participants except for Participant 4 (Table 2, Figure 2).

### 3.5. Changes in miRNA According to the Frozen Period of Breast Milk

#### 3.5.1. Changes in miRNA According to Bottle Warmer Thawing after One Week of Frozen Storage

To assess the changes in miRNA of breast milk according to the duration of frozen storage, all samples were thawed by the same method in a bottle warmer after frozen storage at −20 °C for one and four weeks. When we compared the expression level of miRNAs between the control (Sample 1) and bottle warmer thawing after one week of frozen storage (Sample 5), we saw that two miRNAs showed significant changes in all participants (hsa-miR-205-5p and hsa-miR-3182). Moreover, hsa-miR-3182 was significantly decreased in all participants except for Participant 8 (Table 2, Figure 2).

#### 3.5.2. Changes in miRNA According to Bottle Warmer Thawing after Four Weeks of Frozen Storage

When we compared the expression level of miRNAs between the control (Sample 1) and bottle warmer thawing after four weeks of frozen storage (Sample 7), we saw that four miRNAs showed significant changes in all participants (hsa-miR-103a-3p, hsa-miR-193b-5p, hsa-miR-103b, and hsa-miR-29a-3p). hsa-miR-103a-3p and hsa-miR-103b were significantly decreased in all participants except for Participant 9. hsa-miR-193b-5p was significantly increased in all participants except for Participant 10 (Table 2, Figure 2).

## 4. Discussion

Direct breastfeeding is the ideal method, but due to various social or medical issues, expressed breast milk feeding is often performed, necessitating the collection, processing, and storage of breast milk [14,15]. Breast milk contains several miRNAs as well as nutrients and immune components. Therefore, our study confirmed changes in the miRNA of breast milk by processing, storing, and thawing according to realistic methods rather than methods used in laboratories. Much of the miRNA in milk is encapsulated in exosomes, and it has been shown that these milk exosome miRNAs can resist a low pH (acidic environment), RNase (an enzyme that degrades RNA), and freeze−thaw cycles at −20 °C [16,17,18]. 

In previous studies, Zhao et al. confirmed a decrease in the amount of miRNA and extracellular vesicles in cow milk according to a microwave treatment (305 W for 32 s) [19]. Recently, it has been suggested that a microwave treatment of breast milk (500 W for 40 s) helps prevent the transmission of infections, such as human cytomegalovirus, but it should be considered that this is accompanied by the destruction of several nutrients and miRNAs in breast milk [20]. Therefore, the Centers for Disease Control and Prevention (CDC) abstains from microwave heating for reasons such as destruction of nutrients in breast milk, although thawing breast milk in a microwave is a convenient and practical approach in the home.

In our findings, there were up to six miRNAs with significant differences in expression in all participants, which were less significant than the thousands of miRNAs contained in breast milk. Therefore, it is difficult to suggest that the various storage and thawing processes previously performed have a great influence on the overall expression of miRNA. However, there was no significant change in the amount of miRNA expressed in breast milk subjected to refrigerated storage for one week. This means that refrigerated storage has little effect on miRNA among the storage methods. Many preterm infants are unable to breastfeed; hence, refrigeration storage of breast milk is a common practice in neonatal intensive care units. In previous studies, breast milk refrigerated at 4 °C for up to 96 h showed no significant difference in pH, osmolality, white blood cell counts, bacterial colony counts, and the concentrations of secretory IgA, protein, total fat, and free fatty acids [21,22]. Taking all factors into consideration, the refrigerated storage of breast milk is an ideal storage method for miRNA, as well as macronutrient and immune components.

Two other common storage methods for breast milk other than refrigeration are frozen storage and room temperature storage. When comparing breast milk stored at room temperature for one week (Sample 3) with breast milk thawed in a bottle warmer after one week of frozen storage (Sample 5), four and two miRNAs, respectively, had a significant expression level difference in all participants. After one week of storage at room temperature (Sample 3), the expression level of hsa-miR-378a-3p decreased in all participants, and hsa-miR-365a-3p and hsa-miR-365b-3p decreased in all participants except for Participant 6. hsa-miR-378a-3p has been found to function as a negative regulator of cytokine production involved in interleukin (IL)-33 production and the inflammatory response, and hsa-miR-365a-3p and hsa-miR-365b-3p have been found to function as negative regulators of interleukin-6 production [23,24]. According to Wang et al., miR-365-3p is a negative regulator of IL-17-mediated asthmatic inflammation [25]. All three miRNAs mentioned above have a negative regulatory role in the inflammatory response; it can therefore be suggested that in the case of long-term room temperature storage, the expression level of miRNAs associated with modulation of the inflammatory response may be reduced in breast milk.

When we compared thawing by bottle warming (Sample 5) and microwave (Sample 6) after one week of frozen storage, we saw that the difference in miRNA expression was greater for microwave thawing. The number of miRNAs showing a significant difference in expression was higher when breast milk was frozen for four weeks (Sample 7) than one week (Sample 5). It can be inferred that the longer the frozen storage, the more the denaturation of miRNA. Additionally, a study by García-Lara et al. confirmed that the fat concentration and energy content decreased with the increase in the freezing time of breast milk [26]. When a bottle warmer was used to thaw breast milk stored frozen for four weeks (Sample 7), the expression levels of hsa-miR-103a-3p and hsa-miR-103b decreased in all participants except for Participant 9. According to Trajkovski et al., hsa-miR-103 targeted the insulin receptor regulator caveolin-1 and was involved in the insulin-signaling pathway [27]. The same study further confirmed that the downregulation of miR-103 enhances the caveolin-1 and insulin-signaling pathway, decreases adipocyte size, and increases insulin sensitivity [27].

The composition and expression of miRNAs in HBM can differ due to various maternal, infant, or other factors. However, in this study, we investigated the change in, and stability of, miRNAs according to various storage and thawing conditions rather than the differences between each miRNA according to such factors. Although not in HBM, previous studies have confirmed the change in miRNAs, similar to our research. In a study by Kupec et al., serum miRNA was not affected by food intake or sample collection times. However, there was a significant difference in expression according to the storage temperature (−80 vs. 4 °C) [28]. Glinge et al. confirmed that miRNA levels were stable for at least 24 h at room temperature in whole blood but were significantly changed after 72 h [29]. In addition, the stability was maintained during short-term storage at −80 °C, but there was a significant change during long-term storage for over nine months.

Our research has some limitations. First, only 10 nursing mothers participated in the study; therefore, the total number of participants was insufficient. Second, in this study, seven storage/treatment processes were compared, but if the storage period and processing time had been further subdivided, the results would have been more detailed. Third, more research is needed on how miRNAs are absorbed through the gastrointestinal tract and their functions after being ingested by infants. A study by Luo et al. found that the concentration of corn miRNAs in corn-fed animals remained stable in serum for seven days, indicating that dietary miRNAs are absorbed and distributed in the circulation [30]. Zhang et al. also revealed that miRNA could be regarded as food and a nutrient [31]. However, Dickinson et al. suggested limitations on the bioavailability of miRNAs orally fed to mice [32]. Snow et al. discussed the ineffectiveness of diet-derived miRNAs [33], and Witwer et al. discussed the limited uptake of plant miRNAs in mammalian blood [34]. Such findings illustrate that additional studies are needed to determine whether each miRNA is well absorbed through the gastrointestinal tract in higher animals, such as mammals, and their function.

In conclusion, it is difficult to suggest that the various storage and thawing processes have a great influence on the overall expression of miRNA. However, in the case of refrigeration storage, miRNA changes were minimal compared to other storage and treatment methods. It was possible to suggest that refrigerated storage may be an ideal method to maintain a state of miRNA that is almost similar to that in direct lactation by minimizing changes in miRNAs compared to other storage methods.

## Figures and Tables

**Figure 1 metabolites-13-00139-f001:**
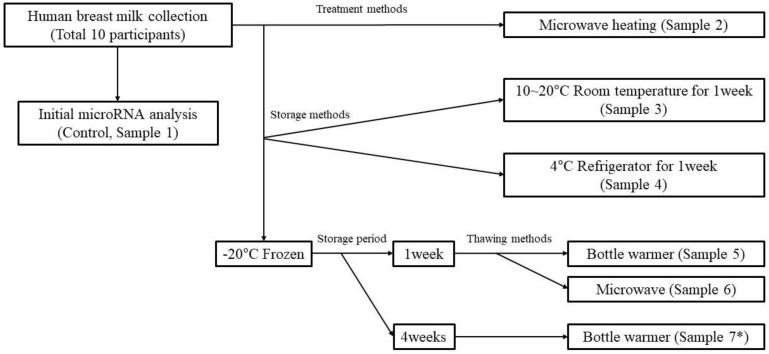
Process of treatment, storage, and thawing of collected breast milk. * Sample 7 treatment procedure was conducted in all participants except for Participant 5.

**Figure 2 metabolites-13-00139-f002:**
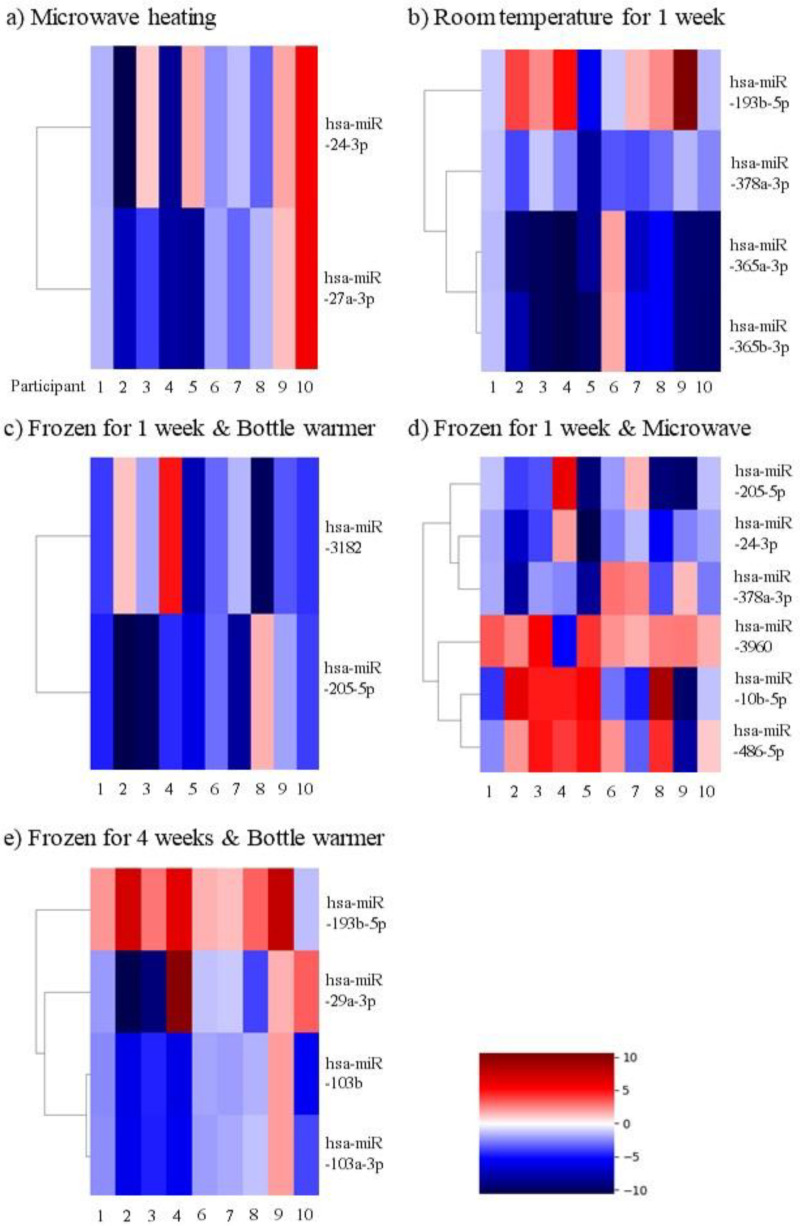
Heatmap of microRNAs showing the significant differences in expression levels compared with the control group.

**Table 1 metabolites-13-00139-t001:** Participant information for microRNA analysis of breast milk.

Participant Number †	Age of Lactating Mother (Years)	Postpartum Periods (Days)	GA at Birth (Weeks)	Birth Weight (kg)	Sex	Delivery Method
1	30	19	38 2/7	3.92	F	NSVD
2	32	70	38 3/7	3.26	M	C/S
3	39	12	24 1/7	0.609	F	C/S
4	28	60	40 3/7	3.65	F	NSVD
5	29	46	34 5/7	2.87	M	NSVD
6	40	146	24 1/7	0.609	F	C/S
7	36	124	40 0/7	3.31	F	C/S
8	40	44	32 2/7	2.12	M	NSVD
9	31	108	38 6/7	2.3	M	NSVD
10	39	217	39 3/7	3	F	C/S
Average	34.4	84.6	35 1/7	2.59		

† Participants 3 and 6 are the same lactating mother who participated in different periods of breastfeeding. GA: Gestational age, NSVD: Normal spontaneous vaginal delivery, C/S: Cesarean section.

**Table 2 metabolites-13-00139-t002:** Fold change of microRNAs with significant expression differences compared to the control group *.

		1	2	3	4	5	6	7	8	9	10
Participants											
Sample 2 Microwaved	hsa-miR-24-3p	0.325	0.001	2.191	0.003	3.250	0.203	0.388	0.106	3.651	52.371
	hsa-miR-27a-3p	0.344	0.007	0.063	0.004	0.003	0.260	0.110	0.343	2.579	53.597
Sample 3 Stored at room temperature (10–20 °C) for 1 week	hsa-miR-193b-5p	0.454	15.714	5.518	34.734	0.019	0.457	2.888	1.262	1526.7	0.349
	hsa-miR-365a-3p	0.361	0.001	0.001	0.001	0.003	3.971	0.008	0.057	0.001	0.001
	hsa-miR-365b-3p	0.382	0.004	0.001	0.001	0.001	3.514	0.019	0.057	0.001	0.001
	hsa-miR-378a-3p	0.416	0.071	0.438	0.165	0.003	0.089	0.072	0.123	0.342	0.169
Sample 4 Refrigerated (4 °C) for 1 week	None **										
Sample 5 Frozen (−20 °C) for 1 week, then thawed in a bottle warmer	hsa-miR-205-5p	0.085	2.175	0.313	19.064	0.011	0.146	0.403	0.003	0.123	0.076
	hsa-miR-3182	0.061	0.002	0.002	0.069	0.028	0.159	0.007	2.651	0.316	0.091
Sample 6 Frozen (−20 °C) for 1 week, then thawed by microwave treatment	hsa-miR-10b-5p	0.065	58.722	21.386	20.856	33.252	0.153	0.047	295.56	0.002	0.442
	hsa-miR-205-5p	0.429	0.076	0.100	51.381	0.003	0.255	2.674	0.003	0.002	0.414
	hsa-miR-486-5p	0.207	4.039	22.914	13.471	24.941	4.238	0.115	16.666	0.006	2.099
	hsa-miR-3960	9.266	5.111	32.091	0.032	15.237	4.287	3.020	5.723	5.989	2.955
	hsa-miR-24-3p	0.304	0.012	0.080	3.575	0.001	0.193	0.403	0.030	0.185	0.282
	hsa-miR-378a-3p	0.324	0.006	0.262	0.200	0.005	6.643	5.442	0.094	2.549	0.171
Sample 7 Frozen (−20 °C) for 4 weeks, then thawed in a bottle warmer	hsa-miR-103a-3p	0.163	0.011	0.032	0.012	***	0.251	0.225	0.297	4.490	0.015
	hsa-miR-193b-5p	5.107	189.26	8.350	115.6	***	3.341	2.711	11.067	339.79	0.370
	hsa-miR-103b	0.176	0.012	0.031	0.014	***	0.227	0.270	0.389	4.607	0.057
	hsa-miR-29a-3p	0.204	0.000	0.001	2065.6	***	0.390	0.442	0.054	3.260	12.458

* Significant expression differences: fold change > 2, normalized data (log2) > 4, *p*-value < 0.05. ** For Sample 4, none of the microRNAs showed a significant change in all participants compared to the control group. *** Sample 7 treatment procedure was conducted in all participants except for Participant 5.

## Data Availability

The datasets generated and/or analyzed during the current study are available in the GEO accession GSE216940 repository. Available online: https://www.ncbi.nlm.nih.gov/geo/query/acc.cgi?acc=GSE216940 (accessed on 5 November 2022).

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
