# Peer review of "Changes in microRNAs during Storage and Processing of Breast Milk"

_metabolites, 2023, doi:10.3390/metabo13020139_

Round 1

Reviewer 1 Report

This is an interesting paper investigating the changes in miRNAs during breast milk storage and processing.  Expressed breast milk samples were collected from the participants and each sample was divided into 7 groups according to the storage temperature, thawing method, and storage period. Several miRNAs were found to be altered in the storage and microwave groups compared with the control group.

1.     Line 54.  Why do you consider mothers aged under 40 years?

2.     Lines 75-77. The statement about milk collection is not very clear.

3.     It is an interesting result that the expression level of the two miRNAs hsa-miR-24-3p and hsa-miR-27a-3p changed between the control and microwave treatment groups in all participants. Does the time of microwave affect the expression level of these two miRNAs?

4.     Line 277. “the Centers for Disease Control and Prevention (CDC) abstains from microwave heating for reasons such as destruction of nutrients in breast milk.” But according to your study, only two miRNAs were changed in the microwave treatment.  However, more miRNAs were altered in other groups compared to the control group. According to this result, do you suggest abstaining from microwave heating?

Author Response

This is an interesting paper investigating the changes in miRNAs during breast milk storage and processing.  Expressed breast milk samples were collected from the participants and each sample was divided into 7 groups according to the storage temperature, thawing method, and storage period. Several miRNAs were found to be altered in the storage and microwave groups compared with the control group.

  1. Line 54.  Why do you consider mothers aged under 40 years?

→ Thank you for your comment. We did not have a particular reason for considering mothers aged under 40. Since information about the subjects is already marked in Table 1, we deleted that part to avoid confusion.

  1. Lines 75-77. The statement about milk collection is not very clear.

→ Thank you for this comment. We agree that the statement requires clarification. We added the following to the methods.

HBM was collected at the Chung-Ang University Breastfeeding Research Institute (Seoul, Korea) for current lactating mothers who agreed to participate in the study. There was no particular restriction on the research participants because our focus was to compare only the change according to storage and thawing with initially collected HBM and confirm the stability of miRNAs.

  1. It is an interesting result that the expression level of the two miRNAs hsa-miR-24-3p and hsa-miR-27a-3p changed between the control and microwave treatment groups in all participants. Does the time of microwave affect the expression level of these two miRNAs?

→ This is an interesting viewpoint. We agree with your perspective. We did not consider how the time of microwave treatment might impact the miRNA. As described in the Discussion section, in previous studies, heating for about 32 or 40 s affects the infection [Zhao Z, et al. Effects of microwave on extracellular vesicles and microRNA in milk. J Dairy Sci. 2018;101(4):2932-2940; Mikawa T, et al. Microwave treatment of breast milk for prevention of cytomegalovirus infection. Pediatr Int. 2019;61(12):1227-1231].

In our study, the milk was heated for 60 s. When heated for more than 60 s, the sample evaporated. Due to the limitation of the sample volume in our study, we could not explore other microwave treatment times, but interesting results could be confirmed if other conditions of the microwave treatment, such as time, were changed with a more sufficient volume. By conducting a follow-up study, we will confirm whether it is possible to obtain a positive effect on infection while minimizing changes to the HBM components, including nutrition or microRNA.

  1. Line 277. “the Centers for Disease Control and Prevention (CDC) abstains from microwave heating for reasons such as destruction of nutrients in breast milk.” But according to your study, only two miRNAs were changed in the microwave treatment.  However, more miRNAs were altered in other groups compared to the control group. According to this result, do you suggest abstaining from microwave heating?

→ The CDC does not recommend microwave treatment because it can destroy the nutrients in milk, without regard for the miRNAs, as explained in the Discussion section, and similar results were confirmed in our previous studies [Kim MH, et al. Macronutrient analysis of human milk according to storage and processing in Korean mother. Pediatr Gastroenterol Hepatol Nutr. 2019;22(3):262-269]. However, as mentioned in our response to your previous comment, we will confirm whether it is possible to obtain a positive effect on infection while minimizing changes to the HBM components, including nutrition or miRNA, in a follow-up study.

Reviewer 2 Report

The authors investigate how thawing process and refrigeration/frozen storage could affect micro RNAs expression in human milk. Authors conclude that it is not possible to suggest optimal thawing and storage processes not depleting microRNAs expression, for human milk. Nevertheless refrigeration seems to minimize changes in microRNAs expression.

General impressions:

The topic and aim of this study are of great interest and helpfulness, especially thinking to the daily routine of a Human Milk Bank and the requirements of premature infants. A strong limitation of the study, surely is represented by the number of participants, which does not allow a distinction between those who gave birth prematurely and those with full-term childbirth.

Remarks:

Introduction:

It would be appropriate to add further study/literature about the functions of microRNAs in terms of newborns’ health. In addition, any factors that may affect the presence and/or expression of microRNAs should be added, focusing the attention on those relevant from a clinical point of view.

Methods:

- Number of participants is a limiting factor of this study, not allowing the distinction between term and preterm milk microRNA profile. In fact, it would be interesting to understand if there are basic differences in terms of microRNA between premature and term mothers and how these differences evolve in relation to freezing, cooling and heating methods. From a statistical point of view, increasing the number of clinically interesting differences. However, no statistical results have been shown.

Results:

- Table1. The range of postpartum period in very wide: does the authors think that this could influence in some ways the results? Especially thinking to the sample 1, the starting point. I mean, do you think that his aspect could biases the sample 1 composition?

- paragraph “changes in microRNA according to the thawing method of breast milk”: the changes reported in terms of expression of microRNAs are attributable to the thawing method or the previous freezing of a week?

Discussion:

-How do you explain the differences in terms of expression of only 6 microRNAs?

-Authors may argue, or speculate more on baseline differences in terms of microRNA and their expression. Are there basal differences? And which could be the factors (maternal or other) possibly associated?

Final consideration: In renewing the usefulness of the results of that study, I would like to stress that they are only a starting point, useful for further insights.

Author Response

The authors investigate how thawing process and refrigeration/frozen storage could affect micro RNAs expression in human milk. Authors conclude that it is not possible to suggest optimal thawing and storage processes not depleting microRNAs expression, for human milk. Nevertheless refrigeration seems to minimize changes in microRNAs expression.

→ Thank you for your interest in our research and understanding our conclusion that refrigeration is a way to minimize changes in the expression of miRNAs.

General impressions:

The topic and aim of this study are of great interest and helpfulness, especially thinking to the daily routine of a Human Milk Bank and the requirements of premature infants. A strong limitation of the study, surely is represented by the number of participants, which does not allow a distinction between those who gave birth prematurely and those with full-term childbirth.

Remarks:

Methods:

- Number of participants is a limiting factor of this study, not allowing the distinction between term and preterm milk microRNA profile. In fact, it would be interesting to understand if there are basic differences in terms of microRNA between premature and term mothers and how these differences evolve in relation to freezing, cooling and heating methods. From a statistical point of view, increasing the number of clinically interesting differences. However, no statistical results have been shown.

Results:

- Table1. The range of postpartum period in very wide: does the authors think that this could influence in some ways the results? Especially thinking to the sample 1, the starting point. I mean, do you think that his aspect could biases the sample 1 composition?

-Authors may argue, or speculate more on baseline differences in terms of microRNA and their expression. Are there basal differences? And which could be the factors (maternal or other) possibly associated?

Final consideration: In renewing the usefulness of the results of that study, I would like to stress that they are only a starting point, useful for further insights.

→ Thank you for your insightful comments. We agree with your comments and viewpoint. We have combined our responses to your comments below.

We were also concerned that the factors, such as the number of participants, prematurity, postpartum periods, delivery mode, and maternal factors, including diet, BMI, and parity, or other factors you mentioned, would lead to baseline differences (sample 1) in collected HBM samples, and that would affect the results and conclusions. This is because these factors can affect the components of breast milk, including immune components, microbiome, and macronutrients. However, it was difficult to confirm that these factors affected the stability of miRNA. Therefore, we focused on whether the miRNA had changed, that is, the stability, rather than confirming the difference according to these baseline miRNAs. In addition, the miRNA change in milk collected from lactating mothers who gave birth prematurely did not show a much different pattern compared to the milk from participants with full-term childbirth.

In the existing research on the stability of serum miRNA, there is no difference according to food intake or sample collection, but storage conditions, especially long-term storage at -80°C (9 months) or room temperature storage for more than 72 h, affected stability. We added to the Discussion section (please see the paragraph at the end of this response).

In addition, according to your advice, we will seek to confirm the difference between human breast milk miRNAs and stability according to these factors (prematurity, postpartum periods, and BMI) in a separate study with more participants.

 The composition and expression of miRNAs in HBM can differ due to various maternal, infant, or other factors. However, in this study, we investigated the change and stability of miRNAs according to various storage and thawing conditions rather than the difference between each miRNA according to such factors. Although not in HBM, previous studies have confirmed the change in miRNAs, similar to our research. In a study by Kupec et al., serum miRNA was not affected by food intake or sample collection times. However, there was a significant difference in expression according to the storage temperature (-80 vs. 4°C) [28]. Glinge et al. confirmed that miRNA levels were stable for at least 24 h at room temperature in whole blood but were significantly changed after 72 h [29]. In addition, the stability was maintained during short-term storage at -80°C, but there was a significant change during long-term storage for over 9 months.

Introduction:

It would be appropriate to add further study/literature about the functions of microRNAs in terms of newborns’ health. In addition, any factors that may affect the presence and/or expression of microRNAs should be added, focusing the attention on those relevant from a clinical point of view.

→ Thank you for your comment. As per your comment, we added the following to the introduction:

 Milk-derived miRNAs may not only serve as a fingerprint of the mother’s health but also of potential outcomes to the health of the infant receiving the milk [9-12]. Mir-181 and mir-155 are associated with the differentiation of B cells, and mir-17 and mir-92 affect the differentiation and maturation of B and T cells. HBM-derived miRNAs also regulate the proliferation of intestinal epithelial cells, have a preventive effect on atopy, and are key regulators of milk lipid metabolism.

- paragraph “changes in microRNA according to the thawing method of breast milk”: the changes reported in terms of expression of microRNAs are attributable to the thawing method or the previous freezing of a week?

→ This is an interesting viewpoint. No samples were frozen before collection, and as soon as samples were collected in a relatively fresh state, freezing began under the same conditions. In addition, freezing for 1 week was common in all thawing methods. Therefore, we judged the difference according to the thawing method rather than the previous freezing for 1 week.

Discussion:

-How do you explain the differences in terms of expression of only 6 microRNAs?

→ Thank you for your comment. In our experiment, the number of miRNAs with expression confirmed was 2,588. However, there is no absolute standard for how many miRNA changes can be determined to be more meaningful. For example, it is difficult to conclude that two miRNA expression changes are meaningless and six miRNA changes are meaningful. In addition, when there were four or six expression changes, there were fewer cases of common increase or decrease in all 10 samples. Thus, some individual miRNA changes were thought to be small numbers from an overall perspective. However, because there was no change during refrigerated storage, it was concluded to be a more meaningful storage method than other methods. The following comment was added to the Results section:

In the HBM samples of 10 participants, the difference in expression was confirmed according to the storage and thawing method of 2,588 miRNAs.

Round 2

Reviewer 1 Report

The authors have responded to my comments, and I suggest accepting this manuscript.